# A cluster of Ankyrin and Ankyrin-TPR repeat genes is associated with panicle branching diversity in rice

Giang Ngan Khong[1]*, Nhu Thi Le[1], Mai Thi Pham[1], Helene Adam[2], Carole Gauron[2], Hoa Quang Le[3], Dung Tien Pham[3], Kelly Colonges[1¤], Xuan Hoi Pham[1], Vinh Nang Do[1], Michel Lebrun[1,4], Stefan Jouannic[1,2]*

1 LMI RICE, National Key Laboratory for Plant Cell Biotechnology, Agronomical Genetics Institute, Hanoi, Vietnam, 2 UMR DIADE, University of Montpellier, IRD, Montpellier, France, 3 School of Biotechnology and Food Technology, Hanoi University of Science and Technology, Hanoi, Vietnam, 4 UMR LSTM, University of Montpellier, IRD, CIRAD, INRAE, SupAgro, Montpellier, France

¤ Current address: CIRAD, UMR AGAP, University of Montpellier, INRAE, Montpellier, France
* ngangiang.khong2010@gmail.com (GNK); stephane.jouannic@ird.fr (SJ)

## Abstract

The number of grains per panicle is an important yield-related trait in cereals which depends in part on panicle branching complexity. One component of this complexity is the number of secondary branches per panicle. Previously, a GWAS site associated with secondary branch and spikelet numbers per panicle in rice was identified. Here we combined gene capture, bi-parental genetic population analysis, expression profiling and transgenic approaches in order to investigate the functional significance of a cluster of 6 *ANK* and *ANK-TPR* genes within the QTL. Four of the *ANK* and *ANK-TPR* genes present a differential expression associated with panicle secondary branch number in contrasted accessions. These differential expression patterns correlate in the different alleles of these genes with specific deletions of potential *cis*-regulatory sequences in their promoters. Two of these genes were confirmed through functional analysis as playing a role in the control of panicle architecture. Our findings indicate that secondary branching diversity in the rice panicle is governed in part by differentially expressed genes within this cluster encoding ANK and ANK-TPR domain proteins that may act as positive or negative regulators of panicle meristem's identity transition from indeterminate to determinate state.

## Author summary

Grain yield is one of the most important indexes in rice breeding, which is controlled in part by panicle branching complexity. A new QTL with co-location of spikelet number (SpN) and secondary branch number (SBN) traits was identified by genome-wide association study in a Vietnamese rice landrace panel. A set of four Ankyrin and Tetratricopeptide repeat domain-encoding genes was identified from this QTL based on their difference of expression levels between two contrasted haplotypes for the SpN and SBN traits. The differential expression is correlated with deletions in the promoter regions of

**Data Availability Statement:** All relevant data are within the manuscript and its Supporting Information files.

**Funding:** This research was sponsored by Vietnam National Foundation for Science and Technology Development (https://nafosted.gov.vn/en/) through grant 106-NN.02-2016.60 to GNK, a joined funding by Agropolis Foundation (« Investissements d'avenir » programme (ANR-10-LABX-0001-01), https://www.agropolis-fondation.fr) and Fondazione Cariplo under the reference ID EVOREPRICE 1201-004 to SJ and the CGIAR Research program on Rice (CRP Rice, http://ricecrp.org) to SJ. The funders had no role in study design, data collection and analysis, decision to publish, or preparation of the manuscript.

**Competing interests:** The authors have declared that no competing interests exist.

these genes. Two of the genes act as negative regulators of the panicle meristem's identity transition from indeterminate to determinate state while the other two act as positive regulators of this meristem fate transition. Based on the different phenotypes between over-expressed and mutant plants, two of these genes were confirmed as playing a role in the control of panicle architecture. These findings can be directly used to assist selection for grain yield improvement.

## Introduction

Rice yield is determined by several traits, including the number of grains carried on each panicle [1,2]. This in turn is dependent on the architecture of the panicle, which consists of a series of axes of different successive orders: rachis; primary branches; secondary branches; potentially tertiary branches; and finally spikelets. Since the rice spikelet typically contains a single fertile floret, the number of spikelets usually determines *in fine* the number of grains that can be produced per panicle. In this context, genetic selection of panicle branching traits to improve yield potential is of great importance in breeding programmes. The complexity of panicle branching is determined by two interconnected developmental processes: the number of axillary meristems produced along each of the panicle axes; and the rate of meristem fate transition, which determines whether an axillary meristem keeps its indeterminacy so as to grow into a higher-order branch (secondary or tertiary branch) or whether it becomes determinate and therefore differentiates into a lateral spikelet. Various genes that control these two processes governing panicle architecture were identified through the characterization of mutants and QTLs, shedding light on the signalling pathways and protein interactions involved [1–5].

Proteins with tandem peptide repeats are essential for fundamental biological functions, through their involvement in protein complexes in which they may perform a scaffolding role [6,7]. Among these proteins, the Ankyrin (ANK) domain containing proteins and the Tetratricopeptide repeat (TPR) domain containing proteins serve as multiprotein complex mediators which function in transcriptional regulation, transport, biosynthesis of the photosynthetic system and protein modification in plants. As a consequence, these proteins regulate diverse biological processes including disease response, abiotic stress response and other processes important for plant growth and development [7–10]. Several genes have been characterized that encode ANK- and TPR-containing proteins involved in reproductive development in grasses. The rice *DECREASED SPIKELET 4* gene (*DES4*) encoding a TPR-LRR protein orthologous to BRUSHY1/TONSOKU/MGOUN3 of *Arabidopsis thaliana*, seems to be required for genomic maintenance like its *A. thaliana* ortholog and affects reproductive meristem activity as illustrated by the decreased spikelet number of the mutant [11–16]. The rice *OsSPINDLY* (*OsSPY*) gene, which encodes a TPR domain-containing N-acetyl glucosamine transferase, modulates plant growth and architecture, including panicle architecture, through its activity as a negative regulator of gibberellin (GA) signalling via DELLA proteins [17–18]. ANK-containing proteins that affect inflorescence development and meristem fate control include the transcriptional co-activators ANK BTB/POZ proteins belonging to the BLADE-ON-PETIOLE (BOP) clade [19]. The BOP gene *TRU1* (for *TASSELS REPLACE UPPER EARS 1*) from maize was identified as a direct target of the TEOSINTE BRANCHED1 (TB1) transcription factor that regulates plant architecture through repression of axillary meristem outgrowth [20]. The loss of function mutant *tru1* is characterized by the presence of long branches tipped by a tassel-like inflorescence in place of the short female axillary branch or ears. The *TRU1* gene may

also play a role in specification of basal leaf compartments, as observed for BOP genes from barley, the rice crop *O. sativa* and the wild rice *Oryza longistaminata* (*CUL4*, *OsBOP1/2/3*, *OlBOP1/2/3* genes respectively) [21–23]. The *CUL4* and *OsBOP* genes regulate tillering and spikelet organ development [21,22], as do their orthologs in *Brachipodium distachyon*, *BdUNICULME4* and *BdLAXATUM-A* [24,25].

To resolve the individual contributions of the different morphological components to spikelet number per panicle (SpN) in rice, we previously performed a genome-wide association study (GWAS) on a panel of Vietnamese rice landraces leading to the identification of 29 QTLs [26]. One of these QTLs on chromosome 2, namely QTL_9, co-associated with SpN and secondary branch number (SBN) traits. In order to validate the GWAS-derived QTL and to identify the gene(s) of functional importance that it represents, a detailed analysis of the corresponding genomic region was carried out in the present study, leading to the identification of two genes affecting panicle architecture within a cluster encoding several different ANK and ANK-TPR containing proteins. Promoter sequence variations, involving deletions of putative transcription factor binding sites, correlate with the differential expression levels of the two key *ANK/ ANK-TPR* genes with respect to panicle secondary branch number.

## Results

### Genetic validation of the GWAS-derived QTL_9

In a previous study, we identified a GWAS site on chromosome 2, designated as QTL_9, that displayed co-localization for the characters spikelet number per panicle (SpN) and secondary branch number per panicle (SBN) [26]. This region of 783 Kbp (positions 16571984 to 17355751) includes 77 annotated coding genes and 3 transposable elements in the *O. sativa* cv. Nipponbare reference genome (Figs 1 and S1 and S2 Table). Among the 24 distinct haplotypes in this region, two main haplotypes with contrasting phenotypic values for the two morphological traits were identified: H1 with low branching values and H2 with high branching ones (S2 Fig). These two haplotypes were selected as they characterized a large number of accessions and a wide genetic distance, while being mainly constituted by *O. sativa ssp. indica* accessions; H1 being *indica* specific. For further genetic analysis, 7 and 5 *indica* accessions were selected respectively from the H1 and H2 haplotype panels, which originated mainly from the Red

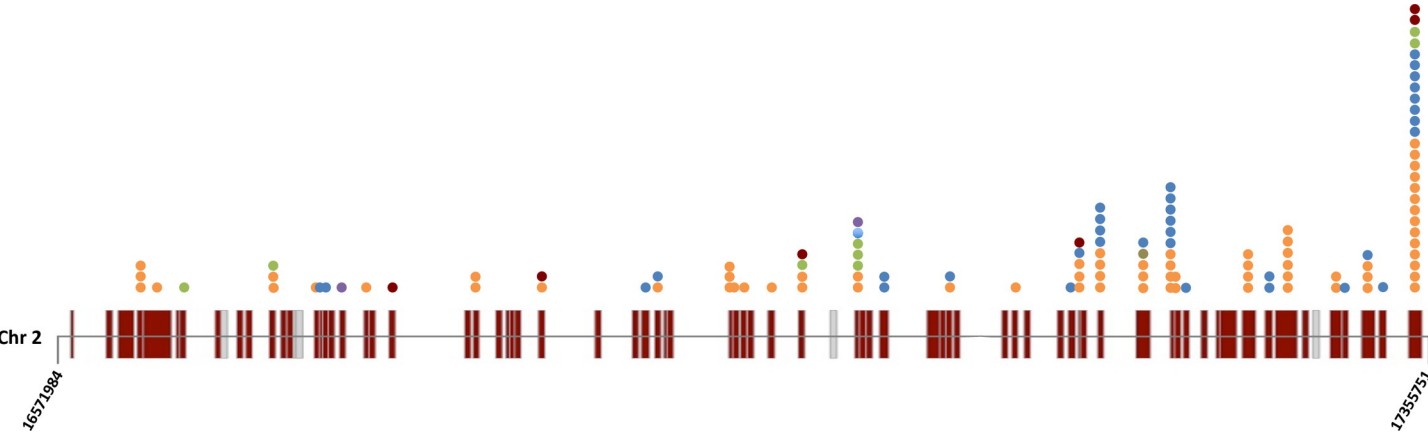

**Fig 1. Polymorphisms between haplotypes H1 and H2 in the QTL_9 intragenic regions.** Schematic view of the QTL_9 region with annotated genes. The genes which were captured for sequencing are indicated in brown and the non-captured genes in grey. The different polymorphic sites identified by gene capture within the genes between the haplotypes H1 and H2 are indicated by coloured dots, with a dot per polymorphic site: orange for non-synonymous coding, blue for synonymous coding, green for frame shift, purple for codon deletion, deep red for Stop gain and light brown for Start gain.

River delta in the case of the H1 accessions and from the North Vietnam mountainous areas for those of H2 (S3 Fig and S1 Table).

A gene capture experiment, based on these selected accessions from H1 and H2 haplotypes, enabled the sequencing of 72 of the 77 genes (including upstream and downstream regions), leading to the identification after filtering of 1035 polymorphic sites between the two haplotypes, including both SNPs and INDELs (S2 and S3 Tables). In the captured intragenic regions, several modifications that entailed protein changes were observed at a higher density towards the 3' end of QTL_9 (Fig 1 and S2 and S3 Tables).

In order to validate the GWAS site, a bi-parental population was produced by crossing the low branching Sớm Giai Hưng Yên (G6) accession with the high branching Khẩu Nam Rinh (G189) accession. These two accessions carry the H1 and H2 haplotypes respectively, which are characterized by contrasting phenotypes for the SpN and SBN panicle morphological traits (Figs 2 and S3). F2 plants (n = 275) were genotyped using CAPS markers derived from Gene Capture-based SNP calling, leading to the selection of 49 homozygous plants over the QTL_9 region for both haplotypes. The phenotyping of the F3 plants revealed significantly divergent values between the two haplotypes for the highly correlated SpN and SBN traits (Figs 2 and S4A), confirming the importance of this genomic region in panicle branching. Phenotyping for other traits revealed no significant segregation for the character primary branch number per panicle (PBN); nor for flowering time, tiller number, or efficient tiller number (S4B and S4C Fig).

## Identification of a cluster of ANK and ANK-TPR genes differentially expressed in QTL9

Of the 77 annotated genes in the *O. sativa* cv. Nipponbare reference genome in QTL_9 region, 16 were reported as expressed during early stages of panicle development according to publicly available databases and panicle-derived RNA-seq data (Fig 3A and S2 Table) [27,28]. In order to determine whether the difference of branching phenotype between the two haplotypes might be associated with differential expression of genes during early panicle development, expression profiling of the 16 genes from QTL_9 expressed in the panicle was carried out for the 2 haplotypes. Panicle developmental staging was defined as follows: "early branching" (from inflorescence meristem stage to panicle with elongated primary and higher order branch development); and "late branching" (from panicle with elongated primary and secondary branches to young flowers with differentiated organs), the first being enriched in meristems of an indeterminate state and the second one with meristems of a determinate state. The expression of the *LOC_Os02g28140* gene in panicle was not confirmed in our condition. Although expression levels often varied between different stages of panicle development, no significant expression differences were observed between the two haplotypes for most of the genes investigated (S5 Fig). However, in a specific region of QTL_9, corresponding to a cluster of 6 genes encoding ANK- and ANK-TPR-containing proteins and 10 additional genes, significant differential expression was observed between the two haplotypes for 4 genes: *LOC_Os02g29040*, *LOC_Os02g29160*, *LOC_Os02g29190* and *LOC_Os02g29210* (Fig 3B and 3C). Two types of branching-related expression profile were observed: higher expression in accessions from the low branching value H1 haplotype (*LOC_Os02g29160* and *LOC_Os02g29210*); and higher expression in accessions from the high branching value H2 haplotype (*LOC_Os02g29040* and *LOC_Os02g29190*) (Fig 3C).

Due to the differential expression between the two haplotypes of certain genes located in the abovementioned gene cluster in the 3' proximal region of QTL_9, we focused on this region in subsequent investigations. The cluster is composed of 6 *ANK* genes and 10 additional

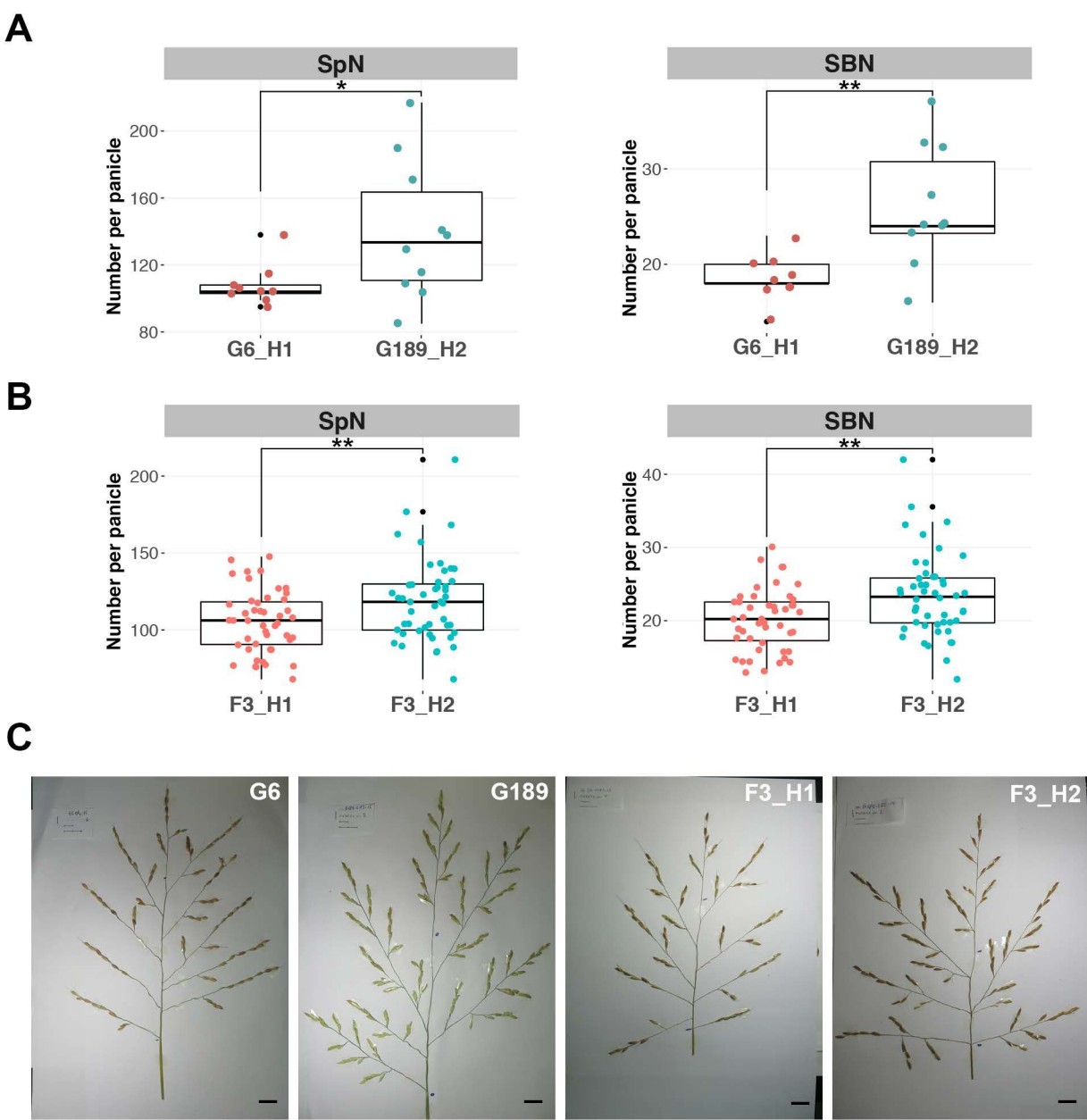

**Fig 2. Panicle architecture in the F3 G6xG189 bi-parental population.** (A) Box-plot with individual dots indicating spikelet number (SpN) and secondary branch number (SBN) per panicle in the two parental accessions G6 from haplotype H1 and G189 from haplotype H2. (B) Box-plot with individual dots indicating spikelet number (SpN) and secondary branch number (SBN) per panicle in the F3 lines from the G6xG189 bi-parental population with H1 haplotype (F3_H1) or H2 haplotype (F3_H2) in the QTL_9 region. (C) Mature panicle from G6 and G189 accessions, a F3_H1 line and a F3_H2 line. Individual dots in box-plots correspond to average values from the 3 main panicles per plant. Scale bar: 2 cm. Statistical significance (*t* test *p* values) between the two lines or parents for the two panicle morphological traits is indicated as follows: * if *p*-value <0.05, ** if <0.01, *** if <0.001.

genes encoding several unknown expressed proteins, a protein kinase and a F-box protein, as well as a short *ANK* gene (*LOC_Os02g29110*) considered to be non-functional (Fig 3B). Five of the 6 *ANK* genes are annotated as belonging to the ANK-TPR subfamily and the last one (*LOC_Os02g29040*) as belonging to the ANK-M subfamily (i.e. proteins with ANK domains only) [8] (S6 Fig). Interestingly, the *LOC_Os02g29040* gene might be derived from an ancestral

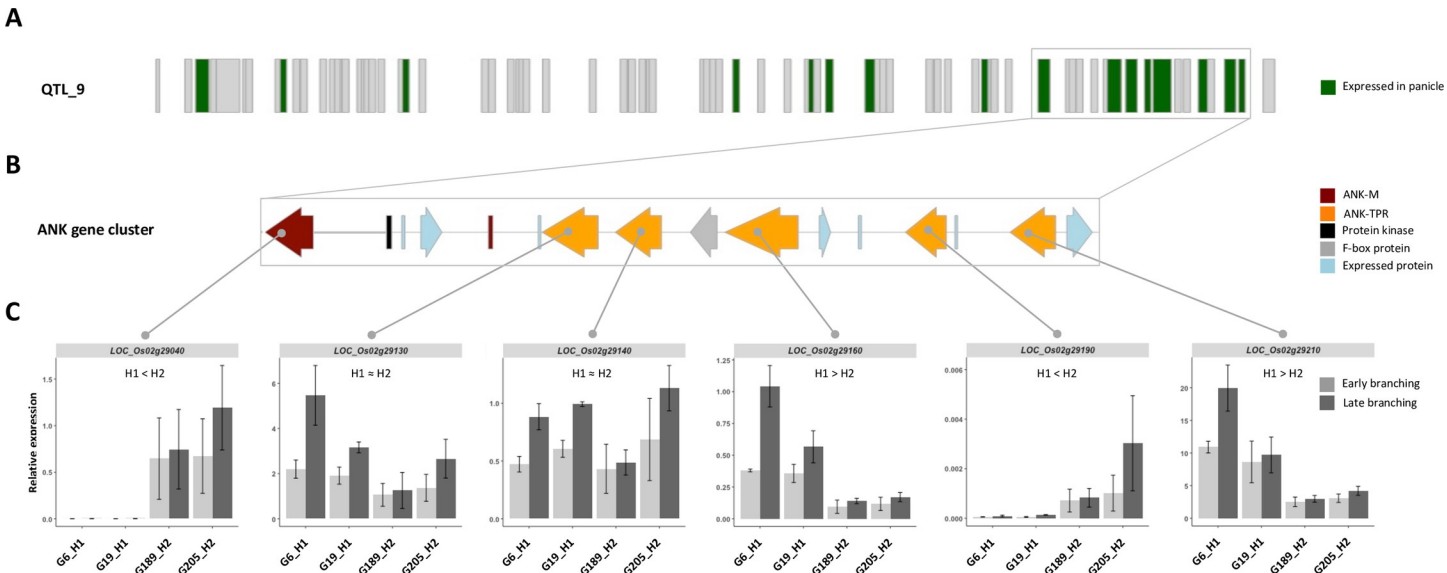

**Fig 3. Structure of the *ANK* and *ANK-TPR* gene cluster in QTL_9 and expression profiling. (A)** Schematic view of the annotated genes according to the *O. sativa ssp. japonica* cv. Nipponbare MSU7.0 reference genome in the 780 Kbp region corresponding to QTL_9 showing genes that are expressed (green) or not expressed (grey) in the developing panicle according to publicly available databases and RNA-seq datasets. **(B)** Schematic view of the cluster of ankyrin genes in the 3' proximal region of QTL_9 with annotated genes shown based on the *O. sativa ssp. japonica* cv. Nipponbare MSU7.0 reference genome. The orientation of the genes is indicated by arrows. **(C)** Histograms of expression profiling by qRT-PCR of the 6 *ANK* and *ANK-TPR* genes in two accessions from H1 haplotypes (G6 and G19) and two accessions from H2 haplotypes, based on two panicle developmental stages: "early branching" and "late branching". Data are given as means ± SD.

*ANK-TPR* gene due to a mutation leading to a premature STOP codon, given that as a coding sequence for TPR domains is present in the long 3'UTR region (S7 Fig). This STOP codon, which is present in other *japonica* and *indica* cultivars, is also observed in the genome of the wild relative species *Oryza rufipogon* (acc. W1943 from Gramene database) (S5 Table). Moreover, sequencing of the *LOC_Os02g29210* cDNA showed that exon 2, as reported in the *O. sativa ssp. japonica* cv. Nipponbare MSU7.0 reference genome, is intronic in the Nipponbare and Kitaake cultivars that we used (S7 Fig).

The differential expression profiles of the *ANK* and *ANK-TPR* genes do not show any relationship that might reflect localization within the gene cluster (Fig 3B and 3C) or molecular phylogenetic similarities (S6 Fig). Indeed, phylogenetic analysis of the *ANK-TPR* subfamily genes from *O. sativa* (17 genes), *Zea mays* (4 genes) and *A. thaliana* (1 gene) revealed that the rice genes clustered in QTL_9, including *LOC_Os02g29040*, are closely related both to each other and to two other rice genes, *LOC_Os03g42350* and *LOC_Os08g13640*. Thus, the gene cluster characterised in the present study is a result of several events of tandem duplications of an ancestral *ANK-TPR* gene. The closest characterised relative of the aforementioned rice genes outside the genus *Oryza* is the maize gene *GRMZM2G536120* (S6 Fig). The unique *A. thaliana ANK-TPR* gene *At3g04710/AtTPR10* (or *AtTPR071* in [10]) is more closely related to the rice gene *LOC_Os05g01310* located on chromosome 5 (S6 Fig).

### *ANK* and *ANK-TPR* gene cluster polymorphisms

Gene Capture data revealed a high degree of polymorphism between the two haplotypes within and around the *ANK* and *ANK-TPR* genes (Figs S8 and 1 and S3 Table). Polymorphisms between the two haplotypes affecting encoded protein sequences were identified in each of the *ANK* and *ANK-TPR* genes in the cluster with the exception of *LOC_Os02g29130* (S9 Fig). Some of these polymorphic sites are non-synonymous. Of these substitutions, 4 are located in

ANK domains and 6 in TPR domains (S9 Fig). Two of the ANK domain polymorphisms might have a functional significance as they occur at well conserved positions involved in the determination of protein structure in LOC_Os02g29210 (T to N in H1 haplotype) and LOC_Os02g29190 (G to E in H1 haplotype) (S9 Fig) [29,30].

All polymorphic sites identified in the coding sequences were also observed to occur within the *indica* subpopulation (n = 1765) from the 3K Rice Genomes project [31], in which additional polymorphic sites could be found (S10 Fig). These missing sites were supposed to be monomorphic between the Nipponbare reference genome and the H1 and H2 haplotypes. The density of polymorphic sites was observed to be higher in the 5' upstream regions of the genes that displayed differential expression between the two haplotypes, with large deletions (from 18 to 42 bp) mainly observed in the H1 haplotype compared to the *O. sativa* cv. Nipponbare reference genome (Figs 4 and S8). Polymorphisms within the *ANK* and *ANK-TPR* genes and their promoter regions between Vietnamese haplotypes H1 and H2 were compared to data available for the *indica* subpopulation of the 3K Rice Genomes project (S11 Fig). We observed that the H1/H2 polymorphic sites were also present in the *indica* subpopulation, with the exception of the large INDELs found in promoter regions in both H1 and H2 haplotypes. The

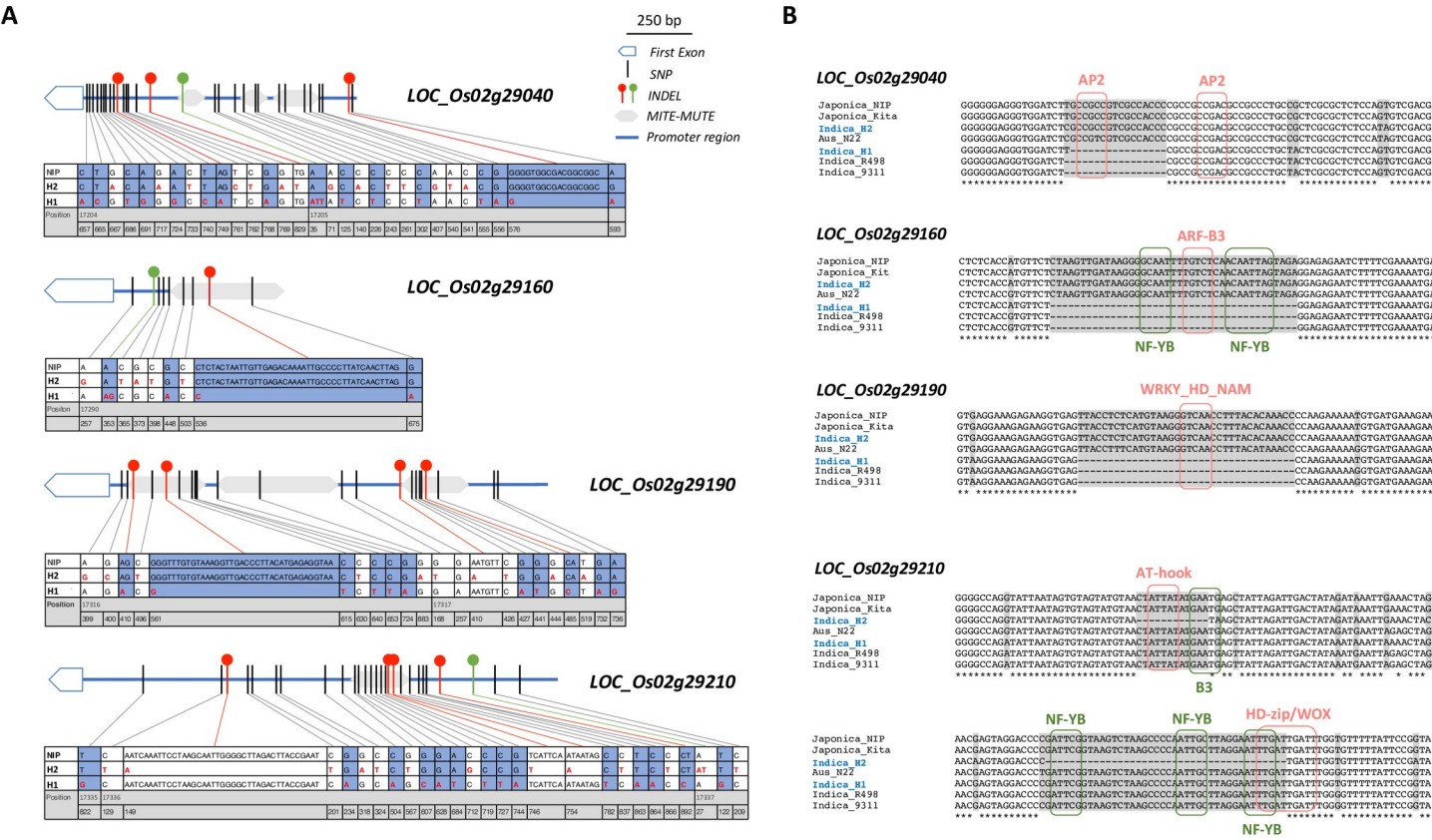

**Fig 4. Promoter regions of the *ANK* and *ANK-TPR* genes with differential expression between the two haplotypes. (A)** Promoter regions and the first exon of the 4 *ANK* and *ANK-TPR* genes showing differential expression between the haplotypes H1 and H2. INDELs are indicated by dotted lines: red with a deletion in H1 and green with a deletion in H2 compared to the *O. sativa ssp. japonica* cv. Nipponbare MSU7.0 reference genome. The sequences and positions of polymorphic sites for the cultivar Nipponbare (Nip) and the H1 and H2 haplotypes (H1, H2) are indicated below, with the H1 vs. Nip polymorphic sites in blue and the H2 vs. Nip polymorphic sites in white. **(B)** Sequence alignment of the promoter regions showing large deletions in the two haplotypes (Indica_H1, Indica_H2) for the *ANK* and *ANK-TPR* genes from the two *O. sativa ssp. japonica* Nipponbare (Japonica_Nip) and Kitaake (Japonica_Kita) genomes; the two *O. sativa ssp. indica* R498 and 93–11 genomes (Indica_R498, Indica_9311 respectively); and the *O. sativa aus* N22 genome (Aus_N22). The positions of the putative TFBSs are indicated by red and green boxes, as well as the names of the corresponding TFs.

latter were found to be generally missing or filtered out in the *indica* subpopulation dataset. The H1 and H2 haplotypes are shared by approximately 16 and 18% of *indica* accessions respectively and were observed to separate widely from each other within our analysis, each clustering with specific *indica* haplotypes from the 3K Rice Genomes subpopulation, to which they are therefore more closely related. Considering the full dataset from the 3K genome project, about 90% and 80% of the accessions sharing H1 and H2 haplotypes respectively are *indica*.

A detailed analysis of the promoter region of the 4 *ANK* and *ANK-TPR* genes revealed that some haplotype-specific deletions had occurred in conserved MULE- and MITE-derived sequences (Figs 4A and S12). Moreover, comparisons with available high-quality full genome sequences from various rice cultivars and from the wild relative *Oryza rufipogon* (W1953 accession) indicated that the large deletions observed in the promoter regions of *LOC_Os02g29040*, *LOC_Os02g29160* and *LOC_Os02g29190* in the H1 haplotype were shared with other *indica* accessions but absent from the wild species (Figs 4B and S12). In contrast, the two large deletions observed in the *LOC_Os02g29210* promoter were not observed in the available full rice genome sequences and might be specific to the H2 haplotype (Figs 4B and S12). More interestingly, the deletions observed in theses 4 genes occurred in predicted transcription factor binding sites (TFBSs) from different families (Figs 4B and S12). From an *O. sativa* ssp. *japonica* cv. Nipponbare RNA-seq dataset obtained previously using a laser microdissection-based approach [27], it was possible to identify several genes which encode TFs of the aforementioned families and which are co-expressed with the *ANK* and *ANK-TPR* genes of interest in panicle reproductive meristems (S13 Fig).

## Functional involvement of the *ANK* and *ANK-TPR* genes from QTL_9 in panicle architecture regulation

In order to determine whether the *ANK* and *ANK-TPR* genes in QTL_9 influence panicle architecture, over-expression and CRISPR-Cas9 genome editing approaches were used to investigate two of these *ANK* and *ANK-TPR* genes, namely *LOC_Os02g29040* and *LOC_Os02g29210*, which show opposing expression patterns between the two haplotypes and which belong to two different subfamilies. The functional analyses were carried out in *O. sativa ssp. japonica* cv. Kitaake.

Three overexpressing lines for the Nipponbare allelic form of *LOC_Os02g29040* gene were selected for phenotyping: ANK1_OX_1, _3 and _4 with median (ANK1_OX_1, ANK1_OX _4) or high (ANK1_OX_3) expression level of the transgene in leaves and panicles (S14A Fig). In the same way, three T1 overexpressing lines for the Nipponbare allelic form of *LOC_Os02g29210* gene were obtained but with low overexpression levels compared to wild-type (S14B Fig). Unfortunately, it was not possible to maintain these plants in normal conditions, since they displayed perturbed growth after transplanting and high sterility, suggesting a strong impact of the ubiquitously expressed transgene on development. In contrast, the 3 overexpressing lines obtained for *LOC_Os02g29040* were characterized by higher PBN and SBN values compared to wild-type, with the exception of the ANK1_OX_4 line which had a similar SBN but a higher SpN value compared to wild-type (Figs 5A, S16B and S16D). These results suggest a positive role for the *LOC_Os02g29040* gene in panicle branching at both levels. The overexpression data corroborate the observation that *LOC_Os02g29040* expression is higher in haplotype H2 accessions, which display higher panicle branching, than in haplotype H1 accessions (Fig 3C).

Using CRISPR-Cas9 technology, it was possible to generate various mutant alleles for the two genes: *ank1-1* to *ank1-3* for *LOC_Os02g29040*; *ank2-1* to *ank2-3* for *LOC_Os02g29210*; plus a double mutant *ank1ank2-1* (S15 Fig). The *ank1-1* allelic form of the *LOC_Os02g29040*

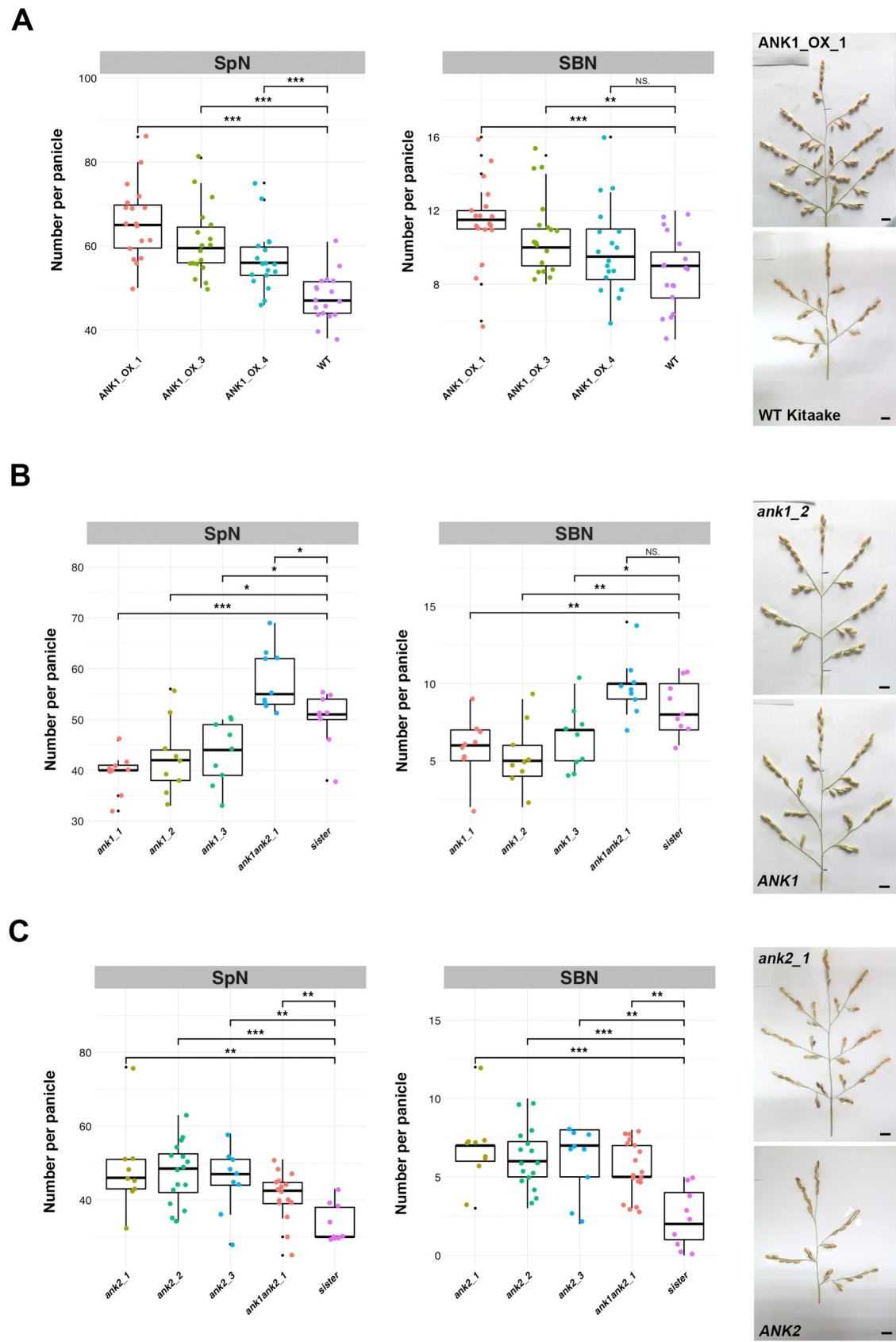

**Fig 5. Involvement of *LOC_Os02g29040* and *LOC_Os02g29210* genes in panicle architecture variation. (A)** On the left, box-plots with individual dots of indicating spikelet number (SpN) and secondary branch number (SBN) per panicle in the different lines overexpressing the *LOC_Os02g29040* gene, in comparison with the wild-type Kitaake cultivar (WT); on the right, mature panicles from the ANK1_OX_1 line and the wild-type. **(B)** On the left, box-plots with individual dots of the spikelet number (SpN) and secondary branch number (SBN) per panicle in the different CRISPR-Cas9-derived lines for the *LOC_Os02g29040* and the double mutant *ank1ank2* (*ank1ank2_1*), in comparison with a "sister" line (i.e. transgenic plant without mutation); on the right, mature panicles from the *ank1_2* line and the "sister" line (*ANK1*). **(C)** On the left, box-plots with individual dots indicating the spikelet number (SpN) and secondary branch number (SBN) per panicle of the different CRISPR-Cas9-derived lines for the *LOC_Os02g29210* gene and the double mutant *ank1ank2* (*ank1ank2_1*), in comparison with a "sister" line (i.e. transgenic plant without mutation); on the right, mature panicles from the *ank2_1* line and the "sister" line (*ANK2*). Individual dots in box-plots correspond to average values from the 3 main panicles per plant. Scale bar: 2 cm. Statistical significance (*t*-test *p* values) between the two lines or parents for the two panicle morphological traits is indicated as follows: NS if the test is non-significant; * if *p*-values <0.05; ** if <0.01; *** if <0.001.

gene encoded a protein with a single amino acid deletion, in contrast to the *ank1-2* and *ank1-3* mutants in which the last 4 ankyrin (ANK) domains were deleted (S15 Fig). The single amino acid deletion in *ank1-1* allele is located at a highly conserved position in the second ANK domain of the LOC_Os02g29040 protein [29]. The *ank2-1* allelic form of the *LOC_Os02g29210* gene encodes a protein with a deletion of 20 aa but with its ANK and TPR domains still conserved. The *ank2-2* mutant encodes a protein with a deletion of the last two ANK domains and all the TPR domains whereas the protein encoded by the *ank2-3* contains a deletion of the third ANK domain (S15 Fig). The double mutant *ank1ank2-1* specified truncated forms of the proteins encoded both LOC_Os02g29040 and LOC_Os02g29210 with only the first two ANK domains (S15 Fig).

The three *ank1* mutants were characterised by normal architecture and growth compared to wild-type. However, in a opposite way of the overexpressing lines, the *ank1* mutants were characterized with a lower number of secondary branches leading to a lower spikelet number, without alterations to either primary branch number or rachis length. The three *ank2* mutants were also characterized by normal architecture and growth compared to wild-type but with a higher number of secondary branches leading to higher spikelet number, without alterations to either primary branch number or rachis length (Figs 5, S16B and S16D). This would suggest a specific role for *LOC_Os02g29040* and *LOC_Os02g29210* in the control of secondary branch number. Compared to *ank1* and *ank2* single mutants, the double mutant *ank1ank2-1* is characterized by higher SpN and SBN values, as seen for the *ank2* single mutants, suggesting a higher impact of *LOC_Os02g29210* mutation on panicle architecture.

Overall, these data indicate that the *LOC_Os02g29040* and *LOC_Os02g29210* genes play a role in the control of panicle architecture in rice.

## Expression profiling of *ANK* and *ANK-TPR* genes from QTL_9 during panicle development

As reported in [8] and in publicly available databases such as Rice-X-Pro (https://ricexpro.dna.affrc.go.jp) or IC4R (http://www.ic4r.org), the rice *ANK / ANK-TPR* genes have different expression profiles in different tissues or organs but generally display a higher expression level in callus, in the panicle, in meristems and/or in spikelets in most cases (S17 Fig). In order to detail their expression patterns during early panicle development (i.e. stages preceding floral organ differentiation), a developmental time course analysis was performed on the Nipponbare cultivar by using a single panicle per stage (Fig 6A and 6B). Genes could be categorised according to their expression levels during two broadly defined phases of development: the indeterminate phase (characterised by inflorescence and branch meristems) and the determinate phase (characterised by spikelet and floret meristems). *LOC_Os02g29040*, *LOC_Os02g29130* and *LOC_Os02g29140* were found to display higher expression during the determinate phase (stages

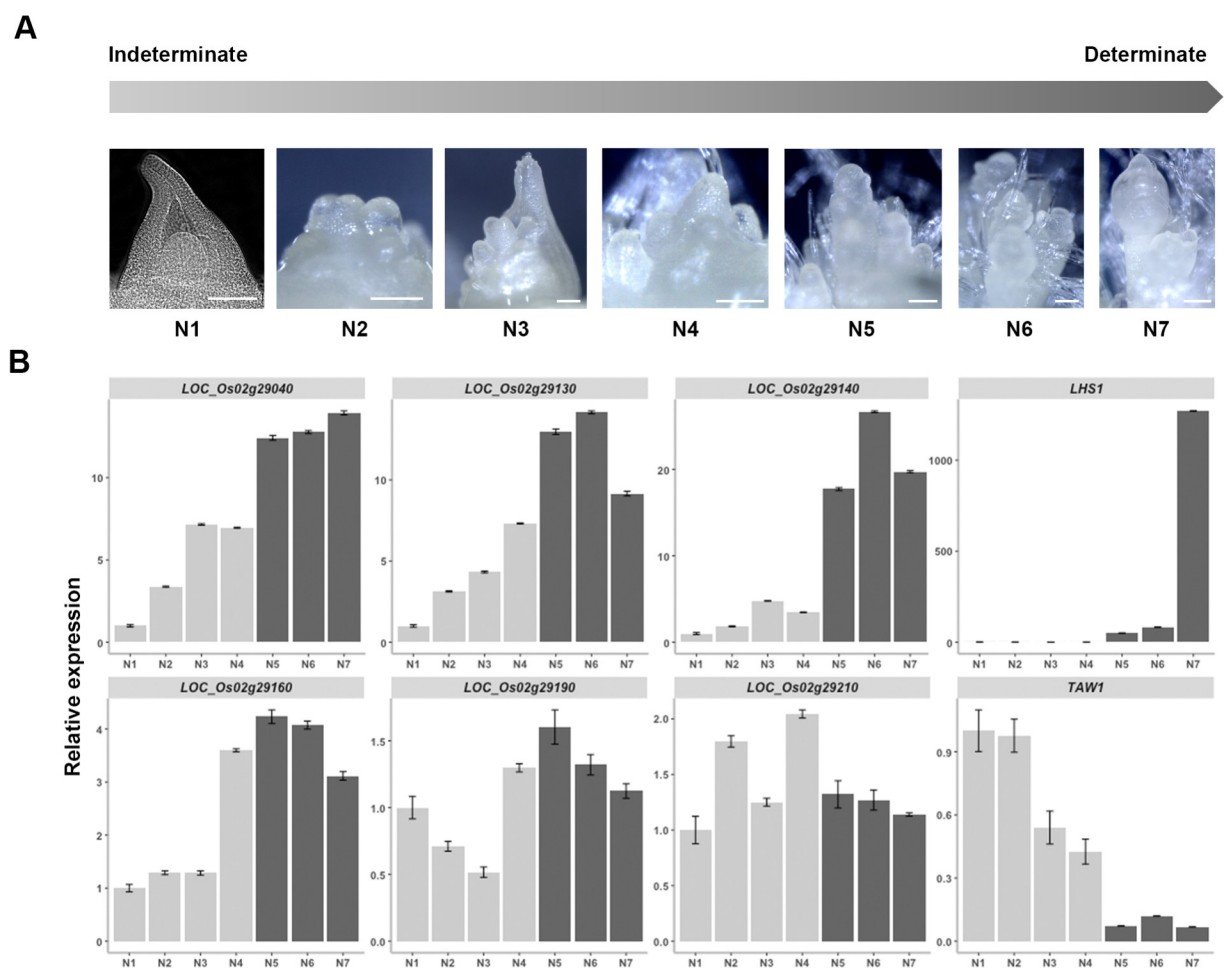

**Fig 6. Expression pattern of ANK and ANK-TPR genes from QTL9 during panicle development. (A)** Panicle staging used for the expression profiling. N1 = rachis meristem; N2 = primary branch initiation; N3 = early primary branch elongation; N4 = primary branch elongation with axillary meristems; N5 = spikelet meristem differentiation; N6 = floret meristem differentiation; N7 = floral organ establishment. Stages N1 to N4 correspond to indeterminate stages, stages N5 to N7 to determinate stages. Scale: 100 μm. **(B)** Expression profiles of the *ANK* and *ANK-TPR* genes during panicle development using a single panicle per stage for Fluidigm qRT-PCR. The *LHS1* and *TAW1* genes were used as quality control of panicle stage sampling. Data are given as means ± SD.

N5 to N7) with a peak at the beginning of spikelet differentiation (stage N5). Similarly, *LOC_Os02g29160* expression was higher from the end of the indeterminate phase until floral organ establishment (stages N4 to N7), with a peak at the spikelet differentiation stage (stage N6). In contrast, *LOC_Os02g29190* and *LOC_Os02g29210* expression levels varied less between the two main phases of the developmental time course in the Nipponbare genetic background. *TAWAWA 1* (*TAW1*) and *LEAFY HULL STERILE 1* (*LHS1*) genes were used as reference expression markers for respectively the indeterminate and determinate phases of the developmental time course [32,33]. Using *in situ* hybridization analysis, transcripts of *LOC_Os02g29040*, *LOC_Os02g29160* and *LOC_Os02g29210* genes were detected in the panicle within both branch and spikelet meristems (S18 Fig).

Overall, these data show that the six *ANK* and *ANK-TRP* genes in QTL_9 in *O. sativa* cv. Nipponbare are expressed during early panicle development in the different panicle meristem types with higher levels at determinate stage (i.e. spikelet and florets) for four of them.

## Discussion

### An ancient *ANK-TPR* gene cluster is contained in the QTL_9 genomic region

Molecular characterization of QTL_9, previously identified by a GWAS study of morphological panicle traits [26], has led us to identify a cluster of ANK- and ANK-TPR-type genes that affect panicle architecture through modulation of secondary branch number and therefore spikelet number per panicle and grain yield.

All six of the QTL_9 *ANK* genes are *ANK-TPR* derived, *LOC_Os04g29040* being an *ANK-TPR* gene with a premature STOP codon that is also observed in the wild species *O. rufipogon*. This protein subfamily shows an expanded number of members in rice compared to other plant species, with 17 genes present in *O. sativa* (in the reference genome MSU7.0), 4 in maize and tomato, and only a single locus in *A. thaliana* [8–10]. The expansion of the *ANK-TPR* gene family in rice has occurred through internal tandem duplications as demonstrated by *ANK-TPR* gene cluster organization in the rice genome [8], especially within the *ANK-TPR* gene cluster in QTL_9. This suggests that an important role has been played by these genes in the evolutionary success of rice through processes of neo- and/or sub-functionalization within the expanded family, leading to enhanced adaptation and morphological diversity. All of the *ANK-TPR* genes from *O. sativa*, as well as the *ANK* gene *LOC_Os02g29040* in QTL_9, are conserved in *O. rufipogon*, indicating ancient duplication events in the Oryza genus that preceded rice domestication. The promoter sequences of these paralogues diverged in parallel to their differential expression patterns within the rice plant as reported here and in previous studies [8,9]. Part of this divergence relates to the differential insertion of *Mutator*-like transposable elements (MULEs) and Miniature Inverted-repeat Transposable Elements (MITEs), which are DNA transposons with high copy numbers predicted to play an important role in genome evolution and gene regulation, notably by providing *de novo* regulatory motifs [34,35]. The detected MULE and MITEs in the promoters of the *ANK* genes from QTL_9 are conserved in various accessions of *O. sativa* in both aus, *japonica* and *indica* ecotypes, as well as in *O. rufipogon*, indicating their ancient origin in the genus *Oryza* before domestication. Nevertheless, the allelic diversity seen in the promoter regions of the *ANK* and *ANK-TPR* genes of QTL_9 has resulted at least in part from deletions within these ancient TEs leading to differential loss of TF binding sites within *indica* accessions after domestication.

### Allelic diversity of the ANK and ANK-TPR proteins

The TPR domain is a 34 amino acid long degenerate repeat corresponding to two anti-parallel α-helices, whereas the ANK domain is a 33 amino acid long degenerated repeat with a structure of two anti-parallel α-helices followed by an anti-parallel β-sheet. Only some of the amino acids in the ANK and TPR motifs are conserved, including hydrophobic positions necessary for the secondary structure [29,30]. Several non-conserved amino acids are responsible for partner molecule recognition and interaction, allowing these domains to bind diverse groups of proteins [29,30]. In this context, it is interesting to note that certain highly conserved amino acids of potential structural and functional importance in the ANK domains encoded by *LOC_Os02g29190* and *LOC_Os02g29210* are altered in *indica* accessions with low branching panicles. In the case of *LOC_Os02g29210*, the presence of potentially deleterious codon changes in the allele found in low branching accessions, coupled with its higher expression level in plants of this type, points to a possible dominant negative role for the encoded protein. Conversely, the putatively non-functional allele of the *LOC_Os02g29190* gene, which is also found in low branching accessions, displays a low expression level, so its impact on branching

may be a result of biological inactivity and/or low expression. Finally, the phenotypic data of the *ank1-1* line carrying a single amino acid deletion in a conserved position of the second ANK domain of the LOC_Os02g29040 protein illustrates the impact of this type of mutation on panicle architecture. In the light of these results, we cannot rule out the possibility that protein variants specified by allelic forms of the *ANK* genes in QTL_9 may influence panicle branching, although the functional effects of the allelic variants remain to be demonstrated.

## The QTL_9 effect is associated with differential expression of genes encoding ANK and ANK-TPR proteins

The importance of gene expression modulation in relation to domestication and morphological diversification in rice is illustrated by several QTLs that affect panicle branching diversity through differential gene expression, as a result of variations in proximal or distal regulatory sequences or via epigenetic post-transcriptional regulation mechanisms, as seen for the cytokinin oxidase/dehydrogenase *OsCKX2* gene, the Squamosa Promoter Binding Protein-Like *OsSPL14* gene, the *ABERRANT PANICLE ORGANIZATION 1* (*APO1*) gene, the gibberellin biosynthesis enzyme *GA20ox4* gene and the AP2/ERF *FRIZZY PANICLE* (*FZP*) gene [36–43]. In the same way, panicle morphological diversity associated with QTL_9 is accompanied by variations in expression levels for 4 out of the 6 *ANK* and *ANK-TPR* genes, two of them (*LOC_Os02g29040* and *LOC_Os02g29190*) showing higher expression in high yield *indica* accessions while the other two (*LOC_Os02g29160* and *LOC_Os02g29210*) display higher expression in the low yield *indica* accessions. These data are corroborated by those obtained with the overexpressing lines for *LOC_Os02g29040* and the loss of function lines for *LOC_Os02g29210*, which in each case produce a higher number of secondary branches and spikelets per panicle.

Expression of the QTL_9 *ANK* and *ANK-TPR* genes was seen to vary in relation to deletions of putative transcription factor binding sites in their proximal regulatory regions. Depending on the target genes, these TFs might act as positive or negative regulators of their expression. The rice *ANK* and *ANK-TPR* genes described here display co-expression during panicle development with several TF genes from the AP2/ERF, WUSCHEL-related homeobox (WOX) and Auxin Response Factor (ARF) families known to be involved in the regulation of inflorescence structure [28,42,44–48]. The latter groups therefore provide good candidates for further investigations of the regulatory network in which the *ANK* and *ANK-TPR* genes operate.

## QTL_9 ANK and ANK-TPR proteins may regulate meristem fate transition during panicle development

Both the higher expression of *LOC_Os02g29040* and *LOC_Os02g29190* and the lower expression of *LOC_Os02g29160* and *LOC_Os02g29210* are associated with a higher secondary branching in accessions of H2 *indica* haplotype. An inverse relationship is seen in lower secondary branched accessions of H1 *indica* haplotype. Consequently, the two groups of *ANK* and *ANK-TPR* genes might act respectively as activators and repressors of secondary branching.

Panicle architecture diversity, notably in terms of spikelet number variation, can be explained at least in part by the fine-tuning of axillary meristem fate on primary branches, which affects the balance between secondary branch (indeterminate) and spikelet (determinate) meristems. In this way, a more highly branched panicle can be obtained by a delayed or lower rate of spikelet meristem fate acquisition [5,49]. The latter hypothesis fits with a general model of inflorescence architecture evolution proposed on the basis of differences in the time period required for terminal and axillary meristems to acquire floral fate [50]. In this context,

the rice *ANK* and *ANK-TPR* genes described here can be separated into two different categories in terms of their influence on meristem fate: either positive regulators of the indeterminate to determinate fate transition leading to lower secondary branching (*LOC_Os02g29160* and *LOC_Os02g29210*); or negative regulators of this transition leading to higher secondary branching (*LOC_Os02g29040* and *LOC_Os02g29190*). According this model, the putatively non-functional allelic form of the *LOC_Os02g29210* gene may act in a dominant negative fashion through binding of a biologically impaired polypeptide to partner proteins that act as negative regulators of the indeterminate to determinate meristem fate transition. On the other hand, the fact that we were unable to obtain overexpressing lines for *LOC_Os02g29210* suggests that drastic, deleterious effects on plant growth and development result when ubiquitous levels of its encoded protein are accumulated.

Since the two types of domain do not themselves have an enzymatic activity or a specific function other than protein-protein interaction, it is likely that the ANK and ANK-TPR proteins are involved in protein complexes, acting as co-chaperone or scaffold proteins. This was recently confirmed for the sole ANK-TPR protein in *A. thaliana* (AtTPR10 or AtTPR071 in [10]) acting *in vitro* as a molecular chaperone in high molecular weight protein complexes with the synergistic involvement of both the ANK and TPR domains [51]. In this context, it can be postulated that the ANK and ANK-TPR proteins in QTL_9 may act as co-chaperones in regulatory multiprotein complexes within panicle meristems.

In conclusion, the wide diversity observed amongst the *indica* rice accessions in our panel, in terms of their panicle architecture, depends in part on four genes encoding ANK- and ANK-TPR-containing proteins that act as positive or negative regulators of the transition from indeterminate to determinate meristem fate during inflorescence development. It will be of great interest to elucidate relationships between the genes described here and others encoding functionally characterized factors involved in panicle development. More specifically, future studies should aim to identify the transcriptional regulators of the rice genes encoding ANK or ANK-TPR proteins, as well as the molecular partners which form protein complexes with the latter within the regulatory network controlling panicle meristem fate. Since the ANK-TPR subfamily in rice is quite large compared to other plant species, it will be important to characterize other members of this subfamily in order to better understand their functional diversity in this species. Moreover, the QTL described in the present study will be of great interest for yield improvement in rice breeding through modulation of panicle architecture to achieve a higher spikelet number.

## Materials and methods

### Gene capture and sequence analysis

Twelve accessions were selected in a collection of *O. sativa* Vietnamese landraces based on GWAS of panicle morphological traits [26]. Seven and five *indica* accessions were respectively selected for the two main haplotypes with contrasting phenotypes: H1 with low panicle branching values; and H2 with high values (S2 and S3 Figs and S1 Table). A set of 80 bp long baits was designed by MYcroarray (www.mycroarray.com), using an overlap of 40 bp between consecutive baits, to target 77 coding genes and upstream regions (excluding transposable elements) located between positions 16574501 and 17363439 on chromosome 2 of *O. sativa* using the MSU v7 reference genome [52] (S2 Table).

DNA extraction and library construction were performed by ANDid (www.adnid.fr) using a home-made kit. Paired end (2x250 bp) sequencing of the 12 libraries was carried out using an Illumina Miseq machine. The initial data analysis was conducted using TOGGLe [53] for read trimming, mapping to the *O. sativa* cv. Nipponbare MSU v7 reference genome and for

SNP and INDEL calling. Annotation of the variants was carried out using the SNiPlay3 web-based application [54]. Using a Minor Allele Frequency (MAF) of 5% and SNPs/INDELs homozygotes, without any missing data for the 12 accessions and discriminating the 2 haplotypes, 1035 variants were investigated (S3 Table).

FASTA sequences from the Vietnamese accessions were deduced manually according to SNP/INDEL calling and the *O. sativa* cv. Nipponbare reference genome. The MBKbase [55] and Rice Genome Hub (https://rice-genome-hub.southgreen.fr) resources were used for sequence comparisons with the 3K genome project [31] and high-quality *O. sativa* genomes. Identification of putative transcription factor-binding sites (TFBSs) was carried out using PlantPAN3.0 website facilities [56] with a cut-off of 0.8 for the "similar score" value to select the *O. sativa* TFBDs. The detection of transposable element sequences was carried out using the RiTE database website [57].

## Development of a bi-parental population for genetic validation

Two *indica* accessions representing the 2 contrasting haplotypes (here after designed as H1 and H2) were crossed: Sớm Giai Hưng Yên G6 (low branching accession from H1 haplotype) and Khẩu Nam Rinh G189 (high branching accession from H2 haplotype). Twenty F1 plants were grown for checking using SSR markers (S4 Table). F2 plants (n = 275) were genotyped using CAPS markers (S4 Table), designed based on Gene Capture-derived SNP calling in QTL_9 region using the CAPSdetector program (https://github.com/francoissabot/CAPSdetector). DNA extractions were carried out using the CTAB method [58]. PCR and restriction enzyme reactions were carried out according to [59]. F3 plants (n = 49 for each haplotype) were grown in lowland field conditions near Hanoi during the 2019 spring season in 0.75 m$^2$ plots of 16 plants each. A block factor of 5 was introduced to check for possible environmental variations within replicates, a single block containing 20 lines (i.e. 20 plots). The three main panicles from three randomly chosen plants per plot were collected (i.e. 9 panicles/accession/replicate). Phenotypic analysis was carried out as described in [26].

## RNA sampling and expression profiling

Panicles were collected from 2 accessions each of the 2 contrasting haplotypes: G6 and G19 for the H1 haplotype; and G189 and G205 for the H2 haplotype (S1 Table). Panicle sampling was carried out by defining two stages: "early branching" (from inflorescence meristem stage to panicle with elongated primary and higher order branch development); "late branching" (from panicle with elongated primary and secondary branches to young flowers with differentiated organs), the first enriched in meristems of an indeterminate state and the second in meristems of a determinate state. Total RNA was extracted using a RNeasy plant mini kit (Qiagen) from two biological replications. For the expression profiling during early panicle development in *O. sativa* cv. Nipponbare, a single panicle per stage was collected for imaging (Leica S8APO stereomicroscope in conjunction with a Leica DFC295 camera) and RNA sampling. Total RNA was extracted using a RNeasy Micro kit (Qiagen) from three biological replications. High-throughput qRT-PCRs using a Biomark HD Microfluidic 96×96 Dynamic Array (Fluidigm) were carried out as described in [28]. RNA probe synthesis and *in situ* hybridization were carried out as described in [60]. All the primers used for expression analysis are listed in S4 Table.

## Plasmid construction and plant transformation

*LOC_Os02g29040* (*ANK1*) and *LOC_Os02g29210* (*ANK2*) CDS flanked by *attB* sequences were isolated from *O. sativa* cv. Nipponbare panicle-derived RTs using nested PCRs

(S4 Table) and cloned into the pC5300 OE vector as described previously [61]. The resulting plasmids (PC5300.OE-*ANK1* and PC5300.OE-*ANK2*) were used for *O. sativa ssp. japonica cv.* Kitaake transformation as detailed in [62]. Single-locus and homozygous T2 lines were selected on the basis of hygromycin resistance and qRT-PCR analysis according to [63], using the single copy gene *SPS* (*Sucrose Phosphate synthase*) as endogenous reference gene [64].

CRISPR/Cas9 plasmid construction was carried out using the polycistronic tRNA-gRNA (PTG-Cas9) method [65]. Two 20 nt gRNAs were designed per gene to target the ankyrin domain-encoding exon 4 in the *LOC_Os02g29040* (*ANK1*) and *LOC_Os02g29210* (*ANK2*) genes (S4 Table). *A. tumefaciens* bacteria carrying the pRGEB32 plasmid with PTG structures were used for the transformation of *O. sativa* cv. Kitaake. CRISPR-Cas9 induced deletions were detected by PCR (see S4 Table for primers) and sequence analysis using DSDecode and CRISP-ID web-based tools [66,67]. T1 Cas9-free plants homozygous for the deletion were selected for further analysis. Phenotyping was carried out on T2 plants using the three main panicles from 6 plants per line as described in [26] and from 6 plants for the other traits (tiller number, panicle number and flowering time).

## Supporting information

**S1 Fig. GWAS-derived QTL_9.** Manhattan-plot and Linkage Disequilibrium (LD) heatmap of chromosome 2 showing significant SNPs (*p*-value threshold $10^{-3}$) in the QTL_9 region for the characters spikelet number (SpN) and secondary branch number (SBN) per panicle for the two field trials performed in 2014 and 2015 according to [26]. Red and bold back lines on the LD heat maps delimit the LD block for the GWAS peak. The significant SNPs are labelled in blue in the LD heatmap. The lower panel shows the annotated genes according to the *O. sativa ssp. japonica* cv. Nipponbare MSU7.0 reference genome within the 783 Kbp region corresponding to QTL_9, indicating genes that are expressed (green) or not expressed (grey) in the developing panicle according the publicly available databases and RNA-seq dataset. (PDF)

**S2 Fig. Haplotype analysis of QTL_9 region in the Vietnamese landrace collection.** (A) Relationship dendrogram of the different haplotypes based on the analysis of the polymorphic sites from QTL_9 region used for the GWAS analysis in the Vietnamese landrace collection [26]. The proportion of *indica*, *japonica* and admixture accessions for each haplotype is indicated. The two main haplotypes, H1 and H2, are indicated. (B) Box-plots of the characters spikelet number (SpN) and secondary branch number (SBN) per panicle evaluated in 2014 and 2015 in the accessions from haplotypes H1 and H2. Statistical significance (i.e. *t* test *p* values) between the two haplotypes for the two panicle morphological traits is indicated in each case. (PDF)

**S3 Fig. Heatmap of the panicle morphological traits in the Vietnamese landrace collection.** Euclidiean_WardD2 heatmap based on the phenotypic values obtained in 2015. Group: genetic group of the accessions according to [68] for *indica* (I1 to I6), *japonica* (J1 to J4) and admixture (m, Im, Jm); Zone: region from Vietnam where the accession was originating (MRD = Mekong River Delta; SE = Southeast; CH = Central Highlands; SCC = South Central Coast; NCC = North Central Coast; RRD = Red River Delta; NW = Northwest; NE = Northeast; u = unknown); Trait types: red for number related traits (PBN, primary branch number; SBN, secondary branch number; SpN; spikelet number), white for length-related traits (RL, rachis length; PBL, primary branch average length; PBintL, average primary branch internode length; SBL, secondary branch average length; SBintL, average secondary branch

internode length). The accessions from haplotypes H1 and H2 are highlighted in blue and green respectively.

(PDF)

**S4 Fig. Characterization of the bi-parental G6xG189 population.** (A) Panicle morphological trait analysis in F3 plants homozygous for the two haplotypes in QTL_9 region. Left panel: Principal Component Analysis (PCA) analysis for the two first axis. Right upper panel: correlation plots of the different panicle morphological traits. Right lower panel: correlation between the SpN and the SBN traits in the two F3 subpopulations (i.e. for haplotype H1 and H2). (B) Box-plots of the panicle morphological trait values in the F3 lines from the G6xG189 bi-parental population with H1 haplotype (F3_H1) or H2 haplotype (F3_H2) in the QTL_9 region: rachis length (RL), average secondary branch length (SBL), average primary branch length (PBL), primary branch number (PBN), secondary branch internode average length (SBintL), average primary branch internode length (PBintL). (C) Box-plots for tiller number (TN), efficient tiller number (eTN), flowering date of the first panicle (FTF), flowering date for 50% of the plants (FT_50). Statistical significance (i.e. *t* test *p* values) between the two haplotypes for the two panicle morphological traits is indicated.

(PDF)

**S5 Fig. Expression profiling of the panicle expressed non-ANK or ANK-TPR genes in QTL_9.** (A) schematic view of the 780 Kbp QTL_9 region showing annotated genes according to the *O. sativa ssp. japonica* cv. Nipponbare MSU7.0 reference genome. Genes expressed in the developing panicle are indicated in green. The *ANK* and *ANK-TPR* genes are indicated by red asterisks. Histograms illustrate the expression profiling data by qRT-PCR of the panicle-expressed genes which do not belong to *ANK* or *ANK-TPR* families in two accessions from H1 haplotypes (G6 and G19) and two accessions from H2 haplotypes (G189 and G205). Two panicle developmental stages were considered: "early branching" and "late branching". (B) Quality control of the panicle sampling. Histograms of expression profiling by qRT-PCR analysis of *LHS1* and *TAW1* genes known to be expressed in late and early stages of panicle development respectively.

(PDF)

**S6 Fig. Ankyrin gene cluster organization, protein structure and phylogenetic relationship.** Phylogenetic tree using the Maximum Likelihood method and JTT matrix-based model in conjunction with amino acid sequence alignment of the ANK domain of the ANK-TPR proteins from *O. sativa* (green dots), *Z. mays* (yellow dots) and *A. thaliana* (blue dot). Bootstrap values (1000 tests) are shown next to the branches. Alignment and phylogenetic tree were carried out using MEGA X software [69]. A schematic view of the organization of the *ANK* gene cluster in QTL_9 based on the *O. sativa* MSU7.0 reference genome is shown on the right of the tree, the grey lines connecting the corresponding genes between the cladogram and the cluster. The orientation of genes is indicated by arrows. The structure of the 6 ANK and ANK-TPR proteins from the cluster is indicated.

(PDF)

**S7 Fig. Annotation of *LOC_Os02g29040* and *LOC_Os02g29210* genes.** The structure of the two genes is indicated as defined in MSU7.0 *O. sativa* reference genome. The sequence of the 3'UTR region of *LOC_Os02g29040* is indicated to illustrate the presence of a TPR domain coding sequence downstream the STOP codon. The sequence of the 5' part of the *LOC_Os02g29210* gene is indicated to illustrate the alternative annotation (i.e. without second exon from MSU7.0 reference genome) of this gene in the Nipponbare accession used in

the lab.
(PDF)

**S8 Fig. Genomic structure and polymorphism of the *ANK* and *ANK-TPR* genes in QTL_9.** Schematic view of the exon-intron structure of the *ANK* genes. Blue box: coding exon; white box: non-coding exon; blue line: intron; black line: promoter region. The structure of the *LOC_Os02g29210* gene was corrected according the *in lab* sequence. Polymorphic sites are indicated with a black vertical line for SNPs and triangles for INDELs. INDEL lengths are indicated in bp. SNPs causing non-synonymous mutations are indicated in red and corresponding variations at nucleic and amino acid levels are indicated.
(PDF)

**S9 Fig. Alignment of the deduced amino acid sequences from the *ANK* and *ANK-TPR* genes coding sequences.** Alignment of the sequences from the *O. sativa ssp. japonica* cv. Nipponbare MSU7.0 reference genome and deduced sequences from the Vietnamese *indica* accessions from the H1 and H2 haplotypes. The ANK and TPR domains are highlighted in green and yellow respectively. A space was introduced between the consecutive TPR domains in the alignment. Modified amino acid in H1 or H2 accessions compared to Nipponbare are highlighted in light green. The positions of non-synonymous polymorphic sites are indicated in grey for amino acid changes within the same aliphatic group and in orange for amino acid changes between different aliphatic groups. The non-synonymous substitutions at highly conserved sites in ANK domain are boxed in red. The region missing from the LOC_Os02g29210 encoded protein sequence deduced from the *in lab* Nipponbare accession (compared with the published Nipponbare reference) is indicated in italics.
(PDF)

**S10 Fig. Polymorphism in the coding sequence of the *ANK* and *ANK-TPR* genes between the two haplotypes in comparison with the *indica* subpopulation from the 3K genome project.** Positions of polymorphic genes are indicated on the schematic view of the gene structure. The sequence and position of polymorphic sites for the cultivar Nipponbare (Nip) and the H1 and H2 haplotypes (H1_Lv, H2_Hv) are indicated below in comparison with the different haplotypes (GID) from the *indica* subpopulation according the data available in the MBKbase website facilities (http://www.mbkbase.org/rice). The number of cultivars for each haplotype is indicated on the right of the table.
(PDF)

**S11 Fig. Promoter regions of the *ANK* and *ANK-TPR* genes with differential expression between the two haplotypes in comparison with the *indica* subpopulation from the 3K genome project.** On the left, promoter regions and the first exon (E1) of the 4 *ANK* and *ANK-TPR* genes showing differential expression between the haplotypes H1 and H2. SNPs are indicated by vertical black lines. INDELs are indicated by dotted lines: red for a deletion in H1 and green for a deletion in H2 compared to the *O. sativa ssp. japonica* cv. Nipponbare MSU7.0 reference genome. The sequence and positions of polymorphic sites for the cultivar Nipponbare (Nip) and the H1 and H2 haplotypes (H1_Lv, H2_Hv) are indicated below in comparison with the different haplotypes (GID) from the *indica* subpopulation according the data available in the MBKbase website facilities (http://www.mbkbase.org/rice). The number of cultivars for each haplotype is indicated on the right of the table. On the right, relationship tree between the different haplotypes using common SNPs in the promoter regions of the *ANK* and *ANK-TPR* genes in conjunction with the Neighbor-joining method.
(PDF)

**S12 Fig. Alignment of the promoter regions of *ANK* and *ANK-TPR* genes displaying differential expression between the two haplotypes.** Sequence alignment of the promoter regions of the two haplotypes (H1_Lv, H2_Hv) of the *ANK* genes with the high-quality full genome sequences from *Oryza rufipogon* (W1953 accession) and rice cultivars: the two *O. sativa ssp. japonica* varieties Nipponbare (Nip) and Kitaake (Kita), the two *O. sativa ssp. indica* varieties R498 and 93–11 and the *O. sativa* aus variety N22. The positions of putative TFBSs are indicated by blue boxes, along with the names of the corresponding TFs. The positions of MULE and MITES-derived regions in the promoter regions are indicated by colour-filled boxes. Polymorphic sites between H1 and H2 accessions are indicated in orange. Polymorphic sites which are monomorphic for H1 and H2 are indicated in light yellow.
(DOCX)

**S13 Fig. Transcription factor-encoding genes that are co-expressed with *ANK* and *ANK-TPR* genes.** Expression profiles of *LOC_Os02g29040* (A), *LOC_Os02g29160* (B), *LOC_Os02g19190* (C) and *LOC_Os2g29210* (D) genes and the co-expressed transcription factor-encoding genes in the laser-dissected panicle meristems according to [27]. M1 = rachis meristem; M2 = primary branch meristem; M3 = elongated primary branch meristem with axillary meristems; M4 = spikelet meristem. The list of TF genes was defined according the list of genes associated with the putative TBDSs (with a similar score ≥ 0.8) based on the search in PlantPAN3.0 website facilities.
(PDF)

**S14 Fig. Expression profiling of the overexpressing transgenic lines for *LOC_Os02g29040* and *LOC_Os02g29210*.** Expression profiling of *LOC_Os02g29040* (A) and *LOC_Os02g29210* (B) in the corresponding overexpressing transgenic lines in comparison with wild-type Kitaake cultivar in leaf and panicle tissues as indicated. Statistical significance (*t* test *p*-values) between the lines and the wild-type is indicated as follows: ns if the test is non-significant, ** if *p*-values <0.01, *** if <0.001.
(PDF)

**S15 Fig. CRISPR-Cas9-induced deletions in *LOC_Os02g29040* and *LOC_Os02g29210* in the single and double gene targeted lines.** (A) Alignment of the nucleic sequence of wild-type Kitaake with the different CRISPR-Cas9-induced alleles. (B) Alignment of the deduced amino acid sequence of the wild-type Kitaake, with those of the different CRISPR-Cas9-induced alleles. (C) Protein structure of the wild-type Kitaake form and the different CRISPR-Cas9-induced alleles.
(PDF)

**S16 Fig. Phenotypic characterization of the CRISPR-cas9 transgenic lines.** (A) Plant architecture at reproductive stages for an ANK1_OE line (i.e. *LOC_Os02g29040* overexpressing line), an *ank1* line and an *ank2* line (CRISPR-Cas9-derived lines for the *LOC_Os02g29040* and *LOC_Os02g29210* genes respectively) in comparison with Kitaake wild-type. (B) Box-plots with individual data points showing primary branch number (PBN) per panicle in the different ANK1_OE lines, in the *ank1* and *ank2* single mutant lines and in the double mutant (*ank1ank2_1*), in comparison with a "sister" line or the wild-type Kitaake cultivar (WT). (C) Box-plots with individual data points showing rachis length (RL) per panicle in the different ANK1_OE lines, in the *ank1* and *ank2* single mutant lines and in the double mutant (*ank1ank2_1*), in comparison with a "sister" line or the wild-type Kitaake cultivar (WT). Statistical significance (i.e. *t* test *p* values) between the two lines or parents for the two panicle morphological traits is indicated as follows: NS if the test is non-significant, * if *p*-values <0.05, **

if <0.01, *** if <0.001.
(PDF)

**S17 Fig. Expression profiling of the *ANK* and *ANK-TPR* genes in *O. sativa* cv. Nipponbare.** Expression profiles of the *ANK* genes in various tissues or organs according to data available in IC4R website (http://www.ic4r.org). TPM: transcripts per million.
(PDF)

**S18 Fig. Spatial localization of *LOC_Os02g29040*, *LOC_Os02g29160* and *LOC_Os02g29210* transcripts in the young developing panicle using *in situ* hybridization.** The histone H4 transcripts (H4) were used as a positive control of the specificity of *in situ* hybridization conditions. SpM: spikelet meristem; BM: branch meristem. Scale: 100 μm.
(PDF)

**S1 Table. List of the Vietnamese landrace accessions used for the Gene Capture analysis.** Zone: Vietnamese region where the accession was originating (MRD = Mekong River Delta; SE = Southeast; CH = Central Highlands; SCC = South Central Coast; NCC = North Central Coast; RRD = Red River Delta; NW = Northwest; NE = Northeast; u = unknown); Group: genetic group of the accessions according to [68] for *indica* (I1 to I6) and admixture (m, Im, Jm); Ecosystem: ecosystem from where the accession originates (RL = rainfed lowland; UP = upland; u = unknown).
(DOCX)

**S2 Table. List of annotated and captured genes in QTL_9.** Relevant information relating to the detection of polymorphic sites within genes and their promoter regions and their impact on protein sequences. Also shown is information on the expression of these genes during early stages of panicle development, based on publicly available data set and local RNA seq dataset. Applicable categories are indicated by green cells.
(DOCX)

**S3 Table. List of filtered SNPs/INDELs from captured genes.** Filtered SNPs and INDELs with a MAF of 5%, homozygotes and polymorphs between the two haplotypes (considering all the tested accessions for each haplotype).
(XLSX)

**S4 Table. List of the primers used in this study.**
(DOCX)

**S5 Table. List of *ANK-TPR* genes from *O. sativa*, *A. thaliana* and *Z. mays*.** The names of *ANK-TPR* genes according to [10] are indicated, as well as the features of the gene and its encoded protein (number of introns, protein, protein length, protein molecular weight, PI). The predicted sub-cellular localization is indicated according to [10]. *O. sativa* genes mentioned in [10] but not maintained in the MSU7.0 version of the *O. sativa* genome are indicated in grey. The presence of *O. rufipogon* orthologs is indicated for the corresponding *O. sativa* genes. *O. sativa* genes from QTL_9 are highlighted in orange.
(DOCX)

## Acknowledgments

The authors would like to thank Laurence Albar (IRD, France) for support with bi-parental population genotyping, François Sabot (IRD, France) for the use of the CAPSdetector program, Brigitte Courtois (CIRAD, France) for providing SSR markers for *indica* genotyping and Céline Cardi and Ronan Rivallan (Grand plateau technique régional de génotypage,

CIRAD, Montpellier) for support with high-throughput qRT-PCR. We also thank James Tregear (IRD, France) for his careful reading and language editing of the manuscript.

## Author Contributions

**Conceptualization:** Giang Ngan Khong, Stefan Jouannic.

**Data curation:** Giang Ngan Khong, Stefan Jouannic.

**Formal analysis:** Nhu Thi Le, Helene Adam, Stefan Jouannic.

**Funding acquisition:** Giang Ngan Khong, Stefan Jouannic.

**Investigation:** Nhu Thi Le, Mai Thi Pham, Helene Adam, Carole Gauron, Hoa Quang Le, Dung Tien Pham, Kelly Colonges.

**Methodology:** Stefan Jouannic.

**Project administration:** Giang Ngan Khong.

**Resources:** Xuan Hoi Pham, Michel Lebrun.

**Supervision:** Xuan Hoi Pham, Vinh Nang Do, Michel Lebrun.

**Visualization:** Stefan Jouannic.

**Writing – original draft:** Stefan Jouannic.

**Writing – review & editing:** Giang Ngan Khong, Stefan Jouannic.

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
