## [Decision Letter · Decision Letter 0]

31 Mar 2021

Dear Dr Jouannic,

Thank you very much for submitting your Research Article entitled 'A cluster of Ankyrin and Ankyrin-TPR repeat genes is associated with panicle branching diversity in rice' to PLOS Genetics.

The manuscript was fully evaluated at the editorial level and by three independent peer reviewers. The reviewers appreciated the attention to an important topic but identified some concerns that we ask you address in a revised manuscript

We therefore ask you to modify the manuscript according to the review recommendations. Your revisions should address the specific points made by each reviewer.

[LINK]

Yours sincerely,

Sarah Hake

Associate Editor

PLOS Genetics

Li-Jia Qu

Section Editor: Plant Genetics

PLOS Genetics

I don’t think you need to do more experiments, but please respond to the reviewer’s comments. I agree with the reviewers that the in situs are not helpful. I am glad you tried them, but they don’t tell us anything. You could put that into supplemental data. I don’t think you need figure 7. You don’t really have enough data to show this without the additional experiments suggested by reviewer 1.

Here are some minor comments.

Make it clear that the ankyrin genes in S5, are different than the ones in figure 3

The section on SNPs is tedious, especially since you don’t actually test any of these SNPs. I think you can try and summarize most of it in a figure rather than writing it out. (For example, you could eliminate lines 193-202). Lines 219-239 could also be shortened.

Figure 4A needs more information. The legend says it is just exon 1, but then what are the different parts? I like to look at a figure and understand it without reading the legend and I am still confused.

Figure 5. what is null sister? Non-CRISPR? Using the word null for non-mutant is confusing.

Lines 383-390 seem unnecessary in the discussion.

Reviewer's Responses to Questions

**Comments to the Authors:**

Reviewer #1: Based on the previous GWAS analysis, the authors performed gene cloning for validating their role in regulating secondary branch and spikelet numbers per panicle, which affects panicle architecture and determines yield. A cluster of 6 ANK and ANK-TPR genes were then analyzed to uncover their putative role in panicle development in this work, including bioinformatics analysis, over-expression and mutant lines phenotype analysis, as well as expression pattern analysis. It’s interesting to find that 4 of them show two haplotypes and play two distinct roles in regulating number of secondary branches and spikelets. Here, I have some suggestions for the authors to revise this paper.

1.There has no direct evidence confirming the correlation between H1, H2 haplotypes and their gene functions, it’s difficult to understand why the authors deduced that the H1 and H2 haplotypes, instead of the other 22 distinct haplotypes in the putative genomic region contribute to the SpN and SBN traits? And is there any evolution clues for H1 and H2 haplotype? Does the MUTE- or MITE- derived sequences in the promoter region help the H2 haplotype evolve from H1 Haplotype? Will the introduction of indica H1 haplotype into japonica background affect panicle development? In Fig 2, the panicles of F3_H1 and F3_H3 are smaller than their parents, does it mean there have other loci contribute to panicle development? To my opinion, we can’t make a definite conclusion without NIL lines analysis between H1 and H2 haplotypes.

2.Regarding to the biological functions of the 6 ANK and ANK-TPR genes, although expression pattern analysis showed that the LOC_Os02g29040, LOC_Os02g29160 and LOC_Os02g29210 have special expression pattern in early panicle development, the in situ hybridization data is not qualified enough to observe any difference between them (TAW1 and LHS1 are suggested to be used for system control). Furthermore, the expression pattern of LOC_Os02g29160 seems quite different to that of LOC_Os02g29210 in Fig.6B. I am afraid the higher expression level at different developmental stages couldn’t explain why the author classified them into two groups. I suggest the author validate their hypothesis by performing promoter swap experiment between LOC_Os02g29040 and LOC_Os02g29210.

3.Following the above question, how about the expression level of LOC_Os02g29210 in the over-expression line of LOC_Os02g29040, as well as in the ank1 mutants? Same to the LOC_Os02g29040, will the expression level of LOC_Os02g29040 change in the ank2 mutants? How about the expression level of other homologous genes in these transgenic lines? And how about the expression pattern of those marker genes, including TAW1, LHS1 and DES4 in these transgenic lines?

4.I am not quite sure about the phenotypes presented in the Fig.5, the wild type Kitaake has the lest number of secondary branches and spikelets than all the other lines? And the panicle of Kitaake seems much smaller than a normal inbreed lines. How many plants and how many panicles were collected for yield traits analysis?

5.There has no direct evidence for validating the regulatory role of LOC_Os02g29160 and LOC_Os02g29190 in panicle development. I don’t think list them in Fig. 7 is correct. Furthermore, in the current model, LOC_Os02g29040 may be a repressor during transition from branch meristems to spikelet meristems, while LOC_Os02g29210 functions as an activator, could the author explain why they have distinct roles in meristem activity.

6. Which generation of ank1 and ank2 were used for phenotype analysis?

7.No figure legends in Fig.7, and in Fig. S5, the icon size are different.

Reviewer #2: The authors present a detailed genetic analysis of a QTL for panicle branching identified through a GWAS experiment. They identify haplotypes within the QTL region that result in high or low branching and high or low spikelet number, and show that these haplotypes have different expression patterns of genes expressed during early panicle development. These genes are characterized by encoding ANK or ANK-TPR proteins that can act in complexes to mediate plant processes. They further create CRISPR mutants of several of the genes encoding these proteins and show that both over expression and knock down has the expected effects on phenotype. They make the claim that these genes mediate the transition from indeterminate to determinate meristems, although I feel that this is too strong an interpretation, and that RNAseq analysis of developing panicles will be needed to find co-regulated genes and to identify what pathways the ANK genes participate in. Some specific questions I have are:

1) the description of the sampling periods ('early' and 'late') for RNA analysis should be explained more fully in the results, as well as its significance. At the moment it is in the M&M, but that puts it well after the Results section.

2) Fig. 6C - it is not clear to me that these illustrate the high and low regulation of these genes, or are specific enough to be clear of their patterns of expression.

**Have all data underlying the figures and results presented in the manuscript been provided?**

Reviewer #1: None

Reviewer #2: Yes

PLOS authors have the option to publish the peer review history of their article (what does this mean?). If published, this will include your full peer review and any attached files.

Reviewer #1: **Yes: **Zheng Yuan

Reviewer #2: No

---

## [Decision Letter · Decision Letter 1]

10 May 2021

Dear Dr Jouannic,

We are pleased to inform you that your manuscript entitled "A cluster of Ankyrin and Ankyrin-TPR repeat genes is associated with panicle branching diversity in rice" has been editorially accepted for publication in PLOS Genetics. Congratulations!

Yours sincerely,

Sarah Hake

Associate Editor

PLOS Genetics

Li-Jia Qu

Section Editor: Plant Genetics

PLOS Genetics

Comments from the reviewers (if applicable):

thank you for your effort in the revision and the response to the reviewers.

Reviewer's Responses to Questions

**Comments to the Authors:**

Reviewer #1: The authors have answered all questions listed by the editor and reviewers, I have no more concerns about this work.

**Have all data underlying the figures and results presented in the manuscript been provided?**

Reviewer #1: Yes

PLOS authors have the option to publish the peer review history of their article (what does this mean?). If published, this will include your full peer review and any attached files.

Reviewer #1: **Yes: **Zheng Yuan

**Data Deposition**

http://datadryad.org/submit?journalID=pgenetics&manu=PGENETICS-D-21-00326R1

**Press Queries**

---

## [Editor Report · Acceptance letter]

3 Jun 2021

PGENETICS-D-21-00326R1 

A cluster of Ankyrin and Ankyrin-TPR repeat genes is associated with panicle branching diversity in rice 

Dear Dr Jouannic, 

We are pleased to inform you that your manuscript entitled "A cluster of Ankyrin and Ankyrin-TPR repeat genes is associated with panicle branching diversity in rice" has been formally accepted for publication in PLOS Genetics! Your manuscript is now with our production department and you will be notified of the publication date in due course.

With kind regards,

Zsofi Zombor

PLOS Genetics

On behalf of:
